# *Cryptosporidium*: Still Open Scenarios

**DOI:** 10.3390/pathogens11050515

**Published:** 2022-04-26

**Authors:** Stefania Pane, Lorenza Putignani

**Affiliations:** 1Department of Diagnostic and Laboratory Medicine, Bambino Gesù Children’s Hospital, IRCCS, Unit of Microbiology and Diagnostic Immunology, Unit of Microbiomics, 00146 Rome, Italy; stefania.pane@opbg.net; 2Department of Diagnostic and Laboratory Medicine, Bambino Gesù Children’s Hospital, IRCCS, Unit of Microbiology and Diagnostic Immunology, Unit of Microbiomics and Multimodal Laboratory Medicine Research Area, Unit of Human Microbiome, 00146 Rome, Italy

**Keywords:** *Cryptosporidium*, zoonosis, humans, animals, transmission, epidemiology, treatment, drug, vaccine

## Abstract

Cryptosporidiosis is increasingly identified as a leading cause of childhood diarrhea and malnutrition in both low-income and high-income countries. The strong impact on public health in epidemic scenarios makes it increasingly essential to identify the sources of infection and understand the transmission routes in order to apply the right prevention or treatment protocols. The objective of this literature review was to present an overview of the current state of human cryptosporidiosis, reviewing risk factors, discussing advances in the drug treatment and epidemiology, and emphasizing the need to identify a government system for reporting diagnosed cases, hitherto undervalued.

## 1. Introduction

In the early 1980s, *Cryptosporidium* spp. was observed as a pathogen affecting both humans and animals and as a cause of acute gastroenteritis, abdominal pain, and diarrhea [1]. Cryptosporidiosis is mainly transmitted via the fecal-oral, zoonotic, or anthroponotic route, through contaminated water or food [2]. Due to the presence of multiple transmission routes, the epidemiology of cryptosporidiosis is complex (Figure 1). Indeed, the investigation of both sporadic cases and outbreaks has contributed to a better understanding of risk factors and infection sources [3]. For example, cryptosporidiosis is increasingly identified as the leading cause of chronic diarrhea in immunocompromised patients and as a cause of infant malnutrition and premature death in children [4]. Despite this, parasitosis is substantially underdiagnosed and underestimated, lacks standardized preventive protocols, and requires optimal treatment on a large scale.

Today, nitazoxanide is the only drug approved by the US Food and Drug Administration (FDA), European Medicines Agency (EMA), and Pharmaceuticals and Medical Devices Agency (PMDA) for the treatment of cryptosporidiosis in immunocompetent people, but its use is not conclusive in young children and immunocompromised individuals [5,6,7], thus having a deep impact on public health today. 

## 2. Epidemiology of *Cryptosporidium*: Recent Updates

Over a million deaths from gastroenteritis associated to *Cryptosporidium* spp. have been described in the last quarter of the 20th century and at the beginning of the 21st century [8]. 

However, the impact of this protozoan parasite remains more widespread in low income countries, where endemicity, poor sanitation, and limited access to filtered water limit diagnostic practice and health care, determining high morbidity and mortality rates, particularly in children under five years of age [9]. The impact of variability in the prevalence of cryptosporidiosis between underdeveloped and industrialized countries makes it difficult to interpret surveillance data and reinforce the necessity to promote prompt diagnosis and improve related methods. The development of 18s rRNA/gp60 gene sequencing has been reported to be useful in the investigation and surveillance of outbreaks and for the identification of unknown species genotypes, reinforcing the understanding of transmission and identifying new key points for effective prevention of infection [10] (Table 1). 

About 44 species of *Cryptosporidium* and more than 120 genotypes [21,58] have been found so far, including *C. hominis* and *C. parvum*, which represent the 95% of cryptosporidiosis cases in humans. *C. meleagridis*, *C. felis*, *C. canis*, *C. ubiquitum*, *C. cuniculus*, *C. muris, C. andersoni*, *C. erinacei*, *C. tyzzeri*, *C. bovis*, *C. suis*, *C. scrofarum*, *C. occultus*, *C. xiaoi*, *C. fayeri*, *C. ditrichi*, *C. viatorum,* and other species are mostly reported in animal infections (https://www.cdc.gov/dpdx/cryptosporidiosis/index.html accessed on 20 May 2019) [11,12,16,18,59,60]. A few cases of *C. meleagridis* infections have also been reported in humans [61], as well as *C. canis, C. felis*, *C. suis*, *C. muris*, and *C. andersoni* in immunocompromised individuals [18,62]. *C. parvum* appears to be the main culprit of zoonotic infections in humans from cattle [14]. However, the absence of host specificity, notably for *C. parvum*, is responsible for infection in multiple hosts, such as mammals, birds, reptiles, and fish [13,63,64]. 

*Cryptosporidium* spp. is transmitted primarily by fecal-oral contamination, via the ingestion of contaminated water or food, or following contact with infected persons or animals [2], although respiratory infections have also been reported [65].

Waterborne cryptosporidiosis outbreaks are the outcome of contaminated municipal and recreational waters, including swimming pools and/or water playgrounds, even in the presence of water treatment plants, because *Cryptosporidium* oocyst stage can withstand disinfections, including chlorination, aiding the parasite to sustain for extended time as viable in the environment [59,66]. In some regions of Africa and Southeast Asia, cryptosporidiosis accounts for 78% of total deaths [60], and is closely related with socio-environmental factors, such as population density, which have a significant impact on the circulation of pathogens [67].

Most developing countries have no governmental system for recording incidence. Furthermore, tests for protozoa are not performed frequently and the prevalence of protozoal infections or water-related epidemics is underestimated, despite the expected high prevalence of water-related parasitic diseases in developing countries (e.g., Latin America, Asia and Africa) where people live in unsanitary conditions [30]. Indeed, there is still no international agreement on the reporting structure of cases, while the surveillance and reporting systems vary widely between the different countries and the comparison of data is not always possible [31].

The contamination of vegetables and fruits with *Cryptosporidium* spp. Has been extensively analyzed and documented [19,32,33,34].

In addition to aqueous and foodborne transmission, *Cryptosporidium* oocysts can be transmitted by the inhalation of aerosolized droplets. Since several studies have shown that respiratory aerosol droplets from infected individuals may represent one of the risk factors in the rapid transmission of *Cryptosporidium* oocysts [68,69,70], it is important that hospital staff use personal protective equipment, especially for the management of immunocompromised patients, to avoid the spread of *Cryptosporidium* oocysts through coughing and sneezing. 

Indeed, oocysts have been detected in the sputum and bronchial aspirates of children with intestinal cryptosporidiosis and cough [71], supporting the hypothesis that *Cryptosporidium* can be transmitted by coughing, sneezing, and expectorate from infected patients.

Considering the different transmission factors and the importance of personal hygiene, *Cryptosporidium* is also responsible for travelers’ diarrhea, which can be considered another source of infection [63].

## 3. Genotyping and Spreading

The identification of *Cryptosporidium* at species and subspecies levels is more and more essential for understanding infection sources and gauging impact on public health in outbreak scenarios. O’ Leary et al. recently analyzed 149 *C. hominis* isolates from Ireland and the United Kingdom using a high resolution fusion melting (HRM) genotyping assay which characterizes and differentiates the six globally prevalent *C. hominis gp60* subtypes, thus providing a profound link between epidemiological cases [8]. 

Several subtypes of *C. hominis, C. parvum*, *C. meleagridis*, *C. tyzzeri, C. cuniculus, C. fayeri,* and *C. ubiquitum* have been identified based on the *gp60* sequence analysis, which is the most widely used genetic marker for subtyping *Cryptosporidium* isolates due to its high degree of genetic divergence [64].

Indeed, reported incidence rates of cryptosporidiosis in Ireland are consistently among the highest in Europe (ranging from 6.58 to 14.17 per 100,000 population, period 2006–2017) [68], with an average of 15 outbreaks annually occurring. The *gp60* gene was sequenced to distinguish *C. parvum* and *C. hominis* subtypes, resulting in an incidence of *C. parvum* species during the spring months, while the incidence of *C. hominis* was confined to late summer and autumnal months. There is experimental evidence to suggest that the IIaA18G3R1 is the dominant subtype infecting cattle and humans in Ireland and Australia. In Portugal, conversely, *C. parvum* IIaA15G2R1 is the most frequent subtype in human and cattle [41,72]. 

Studies conducted in New South Wales, Australia identified *C. hominis* IbA10G2 and *C. parvum* IIaA18G3R1 as the most common subtypes and are equally responsible for the disease [73]. Recent studies suggest an important presence of *C. hominis* in cattle [74,75], which serve as reservoir for the protozoan. Therefore, poor hygiene and intensive farming systems are risk factors for bovine cryptosporidiosis. Epidemiological studies on the presence of *Cryptosporidium* spp. in Italian farms show that the prevalence does not exceed 25% for calves in central Italy [69] or in Sicily, where *C. parvum* was the predominant species [70], while in Sardinian farms the presence of the protozoan was significantly greater (33%) due to much more intensive farming [76].

There are several other *Cryptosporidium* species, including *C. meleagridis, C. canis,* and *C. felis,* that have been reported to be associated with human cases, particularly in developing countries [21]. Indeed, *C. meleagridis* has a broad host range and has been isolated from wild and domestic animals, such as birds, foxes, rats, farm animals, dogs, respectively, and humans. Genotyping studies reported the IIIgA31G3R1 subtype in both poultry and humans, while the IIIbA26G1R1b and IIIbA22G1R1c subtypes have been found in both children and chicken farms in China. In addition, subtypes IIIeA17G2R1, IIIeA19G2R1, IIIeA21G2R1, and IIIeA22G1R1 have been reported in Swedish and Canadian patients as well as in Asian rodents and chickens. *C. canis* and *C. felis* species infect dogs and cats, respectively, but have also been found in humans. In fact, the sequencing analysis of the *gp60* gene has allowed to identify two subtypes (XIXa and XIXb) in *C. felis* and two subtypes (XXa1 and XXa4) in *C. canis*, which were also detected in humans, thus supporting zoonotic transmission [21]. 

Relatively little is known about the epidemiology of zoonotic species of *Cryptosporidium* in captive wildlife. The importance of wildlife in the dissemination of *Cryptosporidium* to drinking water sources and the associated human health risk are still poorly addressed. Indeed, conclusive molecular evidence linking the contamination of water supplies by wild animals as a reservoir of outbreaks of cryptosporidiosis in human populations are scarce. However, pilot studies have reported on waterborne outbreaks link to wildlife origin, such as the case of the UK outbreak caused by *C. cuniculus* from rabbits [77].

A variety of *C. parvum* subtypes, including IIdA15G1, IIdA18G1, and IIdA19G1, isolated from golden takins, lemurs, chipmunks, and hamsters, and IIaA15G2R1, IIaA19G2R1, IIaA19G3R1, IIaA19G4R1, IIaA20G3R1, IIaA20G4R1, IIaA20G3R2, and IIaA21G3R1, from deer and Eastern grey kangaroos has been reported in humans [78,79,80,81]. Other genotypes of *C. ubiquitum*, such as XIIb, XIIc, and XIId [82], from rodents, wild ruminants, carnivores, marsupials, and primates have also been reported sporadically in humans in the US [81].

*C. erinacei* subtypes, including XIIIaA21R11, XIIIaA22R9, XIIIaA21R10, XIIIa20R10, XIIIaA19R12, and XIIIaA22R11, have been identified in hedgehogs, horses, and finally in humans [83]. 

*C. tyzzeri* subtypes, such as IXaA5R2, IXaA6R1, IXaA6R2, IXaA6R3, IXbA6, and IXbA6R2, commonly reported in reptiles, have been occasionally reported in humans [81].

Two studies conducted in Australia identified *C. fayeri* in both immunocompetent [38] and immunosuppressed [37] women. In the first study, *C. fayeri* subtype IVaA9G4T1R1 was identified upon molecular screening and previously identified in Eastern grey kangaroos [84], suggesting a zoonotic transmission. In the second study, the *C. fayeri* subtype IVgA10G1T1R1 identified in the immunosuppressed female patient was previously reported in Western grey kangaroos [85]. Hence, the description of *C. fayeri* clinical infection in human, besides the above reported cases, should be considered a public health concern, overcoming the idea of “host-adapted” species, and reinforcing the idea of zoonotic transmission also for typical wildlife *Cryptosporidium* species. 

The accuracy of genetic analysis has corroborated the understanding of the zoonotic and anthroponotic transmission potential of each species and has improved the knowledge of the epidemiology of *Cryptosporidium* species, leading to advanced strategies for the prevention, surveillance, and control of cryptosporidiosis in humans and other animals [86]. However, one of the limitations of the *Cryptosporidium* subtyping methodology is the false negative rate due to the use of species-specific markers that are able to amplify only *C. parvum, C. hominis,* and related species or genotypes, hence failing to detect the presence of other species potentially present in clinical samples. Furthermore, their applications are still limited due to costs [64]. 

## 4. Geographical Spreading

Only certain countries in the European Union/European Economic Area (EU/EEA) apply *Cryptosporidium* genotyping routinely in surveillance and outbreak investigations, which is an indication of continued single case and outbreak underreporting.

In the European Centre for Disease Prevention and Control (ECDC) report on cryptosporidiosis of 2019, in 2018, the notification rate of reported cryptosporidiosis cases was marginally higher than in the previous four years (2014–2017), with a notification rate of 4.4 per 100,000 population. 

Germany, the Netherlands, Spain, and the UK accounted for 76% of all confirmed cases, with the UK alone accounting for 41%. Notification rates tended to be lower in Eastern Europe than in Western and Northern Europe. Belgium, Finland, Iceland, and the Netherlands report an increase in cryptosporidiosis cases, compared to 2017. The distribution of notified cases increased in April and peaked in September. Furthermore, the highest notification rate was observed in the 0–4 age group, with 17.7 confirmed cases per 100,000 males and 13.7 per 100,000 females. The highest notification rate in this age group was reported from Ireland (87.2 cases per 100,000 population), followed by Belgium (76.3) and the United Kingdom (32.6). Thirteen of the 23 countries for which the rates could be calculated reported less than one case per 100,000 inhabitants in this age group. In Kenya, Mali, Mozambique, Gambia, and South Asia, *Cryptosporidium* infection has caused a higher risk of death in toddlers aged 1–2 years with moderate-to-severe diarrhea (4%) [87].

The male-female incidence rate was nearly equivalent and varied by age group. As in previous years, notifications were higher among boys aged 0–4 years (male-female ratio 1.3/1), as well as among women of reproductive age, with male-to-female ratios of 0.6/1 and 0.5/1 in the 15–24 and 25–44 age groups, respectively. Overall, most of the reported cases (2229) among women in the EU/EEA countries were in the age group 25–44 years. Indeed, advanced genotyping techniques have enabled a fine understanding of the cryptosporidiosis epidemiology in different geographical, seasonal, and even socioeconomic contexts (https://www.cdc.gov/dpdx/cryptosporidiosis/index.html accessed on 20 May 2019).

## 5. Cryptosporidium Treatments

To date, despite the substantial disease burden caused by *Cryptosporidium* spp, treatment options remain limited and only nitazoxanide is approved for treatment, as also demonstrated by experimental studies documenting the inhibition of oocyst excretion [88,89,90].

The studies conducted on the treatment of human intestinal cryptosporidiosis have been different (Table 2). Hussein et al. used drugs such as letrazuril, a chemoprophylactic drug targeting coccidia infections, miltefosine, and clofazimine, which, however, did not show exceptional results on clinical improvement. Paromomycin, in immunocompetent children, also did not show a better effect than nitazoxanide [91].

Even high doses of albendazole, or the use of probiotics or somatostatin analogues showed a substantial reduction in the severity of diarrhea and the number of *Cryptosporidim* oocysts excreted in the faces. A similar result, in the reduction of oocysts, was obtained with the administration of macrolides, such as azithromycin, erythromycin, and roxithromycin [93]. Conversely, the use of acetylated spiramycin in asymptomatic individuals showed a good oocyst reduction rate [92].

Benzoxaborole inhibitors have been identified for protozoal pathogens, appearing to be parasiticidal to *C. hominis* and to potentially inhibit the development of intracellular *C. parvum*. The pyrazolopyridine was tested in neonatal calves and in mice reducing diarrhea and oocyst excretion within three days of treatment [95].

Regarding human trials, in Malawi, protocol trial team members have evaluated the safety, tolerability, pharmacokinetics, and efficacy of clofazimine on HIV-infected individuals with and without *Cryptosporidium* diarrhea, measuring the reduction in oocysts excretion as primary outcome, and non-cessation of clinical symptoms, which is a secondary outcome. The aim of clinical trial is also the future use of clofazimine in children of 6–18 months of age, who so far do not have a definitive treatment [94].

In general, the different therapeutic treatments take into account the clinical improvement defined as a reduction of diarrhea and complete eradication or reduction of oocysts. Several other trials are currently seeking to clarify the protective immune response against *Cryptosporidium* infection for effective vaccine development [95].

Currently, there is not yet a recognized vaccine and the only available pharmacological treatment approved by the FDA, EMA, and PMDA, as discussed, is nitazoxanide. Indeed, the development of new effective pharmacological treatments and vaccines is very slow, due to the lack of efficient *Cryptosporidium* in vitro systems and animal models able to validate in vivo the efficacy of drugs, and due to still prohibitive costs affecting trials [64,96].

These important limitations need to be overcome by access to new treatments, exploiting recent findings and filling gaps in the current clinical knowledge and management of cryptosporidiosis [97].

The use of probiotics is an alternative emerging therapeutic strategy for *Cryptosporidium.* Sindhu et al. reported the effect of the probiotic *Lactobacillus rhamnosus* GG on bowel function, immune response, and clinical outcomes in Indian children with *Cryptosporidium* diarrhea. According to their studies, the probiotic can prevent further damage and promote intestinal integrity in children [98]. Moreover, Pickerd et al. reported the benefits of *Lactobacillus rhamnosus* GG and *Lactobacillus casei* in the treatment of cryptosporidiosis in humans resulting in the timely clinical improvement and resolution of the infection [99]. Furthermore, since probiotics are alive microorganisms, it is believed that they can balance the gastrointestinal microbiota, thus preventing *C. difficile* associated diarrhea (CDAD) caused by antibiotics [100]. 

## 6. Microbiota and *Cryptosporidium* spp.

The human gut is a highly complex ecosystem with an extensive microbial community, and the influence of the intestinal microbiota reaches the entire host organism. Hence, the maintenance of homeostasis between the gut microbiota and the rest of the body is crucial for health [101].

The alterations of the eubiotic composition of the gut microbiota have been investigated in neonatal calves as the result of *C. parvum* infection, showing an increase in *Fusobacterium* abundance, followed by genera belonging to Bacteroidetes, Proteobacteria, Fusobacteria, and Actinobacteria phyla [102]. 

In human, Carey et al., analyzed 72 fecal samples from Bangladesh children to profile the microbiota during cryptosporidiosis. The authors noted that the composition of the microbiota was predictive of diarrheal symptoms both before and during the infection. *Megasphaera* genus was observed at high abundance in cases of subclinical *Cryptosporidium* infection while it was scarce or absent in cases of diarrhea. Interestingly, this observation could suggest that *Megasphaera* may prevent acute diarrhea during the parasite infection or at least can serve as biomarker for still unknown protective factors that could be further investigated [103].

Still, the complex interplay between parasites and the gut microbiota is poorly understood, and there are only few and often conflicting results both in humans and in animal models [104]. In the study of Bednarska et al., an investigation was conducted to determine the prevalence of intestinal pathogens, such as *Cryptosporidium*, *Giardia*, *Blastocystis*, *Cyclospora,* and microsporidia, in hospitalized patients with different immunological statuses. *Cryptosporidium* and *Cyclospora* were diagnosed as the main cause of heavy diarrhea and adult patients were positive mainly for *Blastocystis* and microsporidia, while children were more often for *Cryptosporidium* species [105].

To assess the comparative role of *Cryptosporidium* spp. and other enteric pathogens in animals, a pilot study was conducted in neonatal calves [106]. In this study, neonatal calves were screened for *Cryptosporidium parvum*, *Escherichia coli* K99, Rotavirus, and Coronavirus. Such mixed infections are not uncommon and are usually believed to exacerbate the clinical severity when they occur [107].

In another study, Garro et al. evaluated cryptosporidiosis of neonatal dairy calves in the presence of other enteropathogens, such as Rotavirus of group A, *bovine* Coronavirus, and enterotoxic *E. coli*. *Cryptosporidium* spp. was found to be the main etiological factor of diarrhea in the neonatal calf group, while Rotavirus of group A appeared to play only a secondary role in the etiology of diarrhea, suggesting that these infectious events could be independent. In contrast, mixed infection were virtually absent in older calves [108].

Charania et al. examined changes in the mouse gut microbiota following antibiotic treatment to determine how cryptosporidial infections and gut integrity were affected by alterations of the microbiome. There was a significant decrease in anaerobes and an overgrowth of Enterobacteriaceae in mice treated with cloxacillin. Moreover, there was a significant decrease in acetate, propionate, and butyrate in these same mice. Parallel to the decrease in bacterial infection, a significant increase in the severity of cryptosporidial infection and increase in gut permeability were registered. Treatment with other antibiotics significantly altered the microbiome but did not change the infection, suggesting that specific alterations in the host microbiome allow for a diversified growth of the parasite [109].

Mammeri et al. characterized the impact of *C. parvum* infection on the goat kid microbiome. *C. parvum* was orally administered to parasite-naïve goats, and infection was monitored for 26 days in fecal samples. *C. parvum* decreased the abundance of butyrate-producing pathways in bacteria and increased mucosal inflammation and tissue repair [110].

Clearly, the understanding of the interactions between the intestinal microbiota and enteric pathogens, such as *Cryptosporidium* or other parasites, is of great interest in the development of alternative treatments (e.g., probiotics, prebiotics, synbiotics) not relying on chemotherapy, and for this reason is an emerging area of research [111].

## 7. Study Models

Progress towards development of therapeutics for cryptosporidiosis has been hampered by lack of optimal models to support long-term cultivation of the parasite. 

Several in vitro culture systems have been reported to model *Cryptosporidium* infection [112,113]. Two-dimensional (2D) cultures of colorectal adenocarcinoma cell lines have been most frequently used, but most of these only support short term infection (<5 days) and incomplete propagation of the parasites [114].

For this reason, new organoid technology may represent a powerful and alternative model to be used to culture and propagate *Cryptosporidium* and to understand the host–parasite interactions (Figure 2).

The organoid is an in vitro 3D cell cluster derived from stem cells or organ progenitors that reproduces the general structure of an organ as in vivo. 

Organoids simulate the in vivo situation much better than the intestinal cell lines currently used for studying host–parasite interactions in vitro [87,115], and can support continuous parasite culture for 28 days.

Organoids may enable the identification of genes as possible therapeutic targets to develop new effective therapies. Heo et al. used small intestinal and lung organoids from healthy human donors to model the infection of *C. parvum*. Indeed, differentiated intact 3D intestinal organoids microinjected with sporozoites supported the full replicative cycle, resulting in newly generated oocysts that were infectious to neonatal mice. As observed in small intestinal organoids, *C. parvum* also forms oocysts within lung organoids because they replicate the in vivo conditions of infection [116]. 

## 8. Materials and Methods

The literature search was performed using the PubMed database in combination with the following search terms: *Cryptosporidium*, zoonosis, humans, animals, transmission, epidemiology, treatment, drug, nitazoxanide, vaccine. We searched PubMed original article, case reports, bulletins of CDC and ECDC, and reviews, accessing the literature published prior to 31 December 2021. Moreover, we reported on the cryptosporidiosis in humans and animals by preferring the literature focused on emerging epidemiological and drug treatment data.

## 9. Conclusions

Cryptosporidiosis remains a global health emergency, especially for children and severely immunocompromised patients. Currently, the lack of a government system for recording disease incidence and gold standard identification methods leads to an underestimation of the incidence of cryptosporidiosis. *Cryptosporidium* spp. Is the pathogen most frequently associated with food- and water-borne outbreaks and enteric infections [117]. To better control the spread of this pathogen, an educational prevention program on this parasitosis should be implemented, focusing on hygienic behavior recommendations for individuals. In fact, adopting some simple hygienic procedures, both at home and personally, such as cleaning contaminated surfaces with appropriate disinfectants, boiling tap water, or washing hands, can reduce the risk of spreading the infection. It is also important to apply efficient wastewater treatment processes that can prevent the environmental transmission of *Cryptosporidium*.

Although cryptosporidiosis is a worldwide cause of diarrheal diseases, no antiprotozoal agent or vaccine exists for its effective treatment or prevention. Pharmacological therapies fail to reduce the spread of oocysts to obtain clinical benefits. Considering the limited treatment options, there is therefore a need to develop a safe and effective therapy against cryptosporidiosis to improve the lives of children with acute illness and immunodeficient individuals [118]. In this regard, new disease study models based on organoid platforms have been suggested, which allow to study, in vitro, host–parasite interactions to identify new therapeutic targets.

Thus, the effectiveness of local and national surveillance and progress in drug development remain of paramount importance to contain the epidemiology of this important but often underestimated pathogen. 

## Figures and Tables

**Figure 1 pathogens-11-00515-f001:**
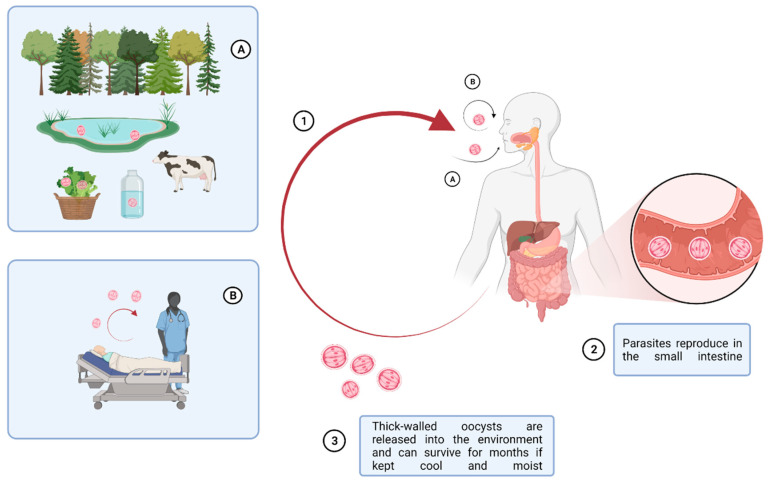
Multiple factors affecting the routes of *Cryptosporidium* spp. infection cycle. Zoonotic *Cryptosporidium* parasites are transmitted from livestock through long-lived oocysts in their faces, which can contaminate the environment, water, and food, producing a source of infection to people. Besides water- and food-borne transmissions, inhalation of oocysts has been described as another mode of transmission. (1) oocyst ingestion by (**A**) fecal-oral route and (**B**) inhalation; (2) parasite reproduction in the small intestine; (3) release of oocysts in the environment. This picture was created with BioRender.com accessed on 20 May 2019.

**Figure 2 pathogens-11-00515-f002:**
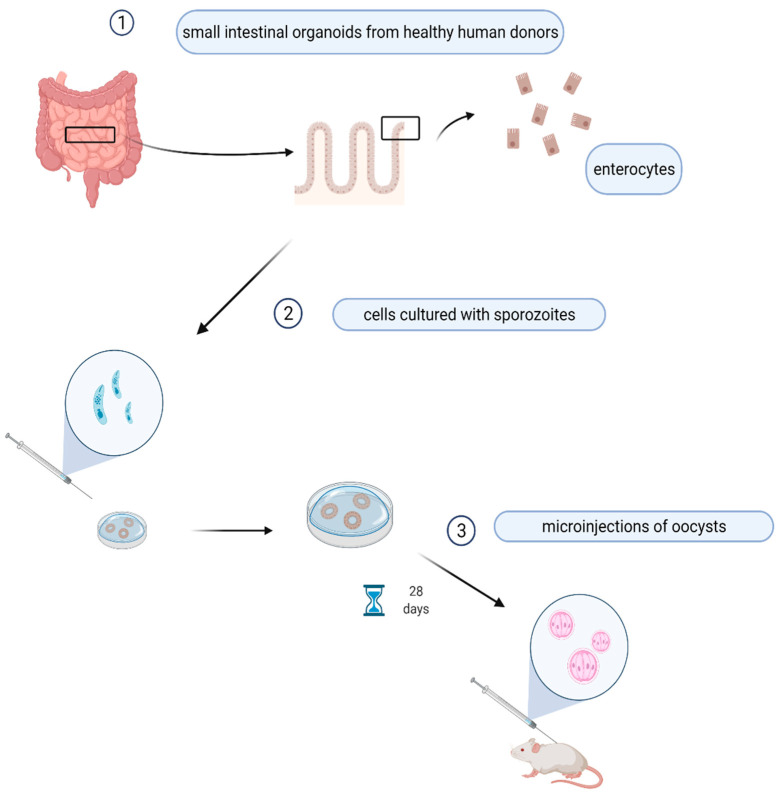
Prototype of organoids’ simulating in vivo growth of *Cryptosporidium* spp. Enterocytes from patient’s intestine are isolated and cultured with sporozoites for 28 days, then the organoides are injected in the mouse. This picture was created with BioRender.com accessed on 20 May 2019.

**Table 1 pathogens-11-00515-t001:** *Cryptosporidium* species identification, confirmed by polymerase chain reaction (PCR) and sequence analyses in animals and in humans.

Species Name	Major Host(s)	Human Host	Author(s)
*Cryptosporidium cuniculus*	rabbits	yes	[11]
*Cryptosporidium hominis*	equine	yes	[12,13]
*Cryptosporidium parvum*	sheep, cattle	yes	[13,14]
*Cryptosporidium muris*	rodents and farm animals	yes	[13,15]
*Cryptosporidium andersoni*	cattle	yes	[13,16,17]
*Cryptosporidium suis*	pigs	yes	[13,18,19]
*Cryptosporidium wrairi*	guinea pigs	yes	[13,16]
*Cryptosporidium meleagridis*	birds	yes	[13]
*Cryptosporidium canis*	dogs	yes	[13,17]
*Cryptosporidium ubiquitum*	ruminants and rodents	yes	[18]
*Cryptosporidium xiaoi*	sheep and goats	yes	[18,19]
*Cryptosporidium felis*	cats	yes	[20,21]
*Cryptosporidium nasorum*	fish	not	[22]
*Cryptosporidium molnari*	fish	not	[23]
*Cryptosporidium scophthalmi*	fish	not	[24]
*Cryptosporidium scrofarum*	pigs	yes	[25,26]
*Cryptosporidium baileyi*	chicken	yes	[27,28]
*Cryptosporidium galli*	chicken	not	[29]
*Cryptosporidium fragile*	amphibia	not	[30]
*Cryptosporidium serpentis*	snakes	not	[31]
*Cryptosporidium varanii*	pet reptiles	not	[32]
*Cryptosporidium ryanae*	cattle	not	[33]
*Cryptosporidium bovis*	cattle	yes	[34,35]
*Cryptosporidium fayeri*	marsupials	yes	[36,37,38]
*Cryptosporidium macropodum*	marsupials	not	[39]
*Cryptosporidium tyzzeri*	mice	yes	[40,41]
*Cryptosporidium viatorum*	rodents	yes	[42]
*Cryptosporidium occultus*	rodents	yes	[43]
*Cryptosporidium proventriculi*	birds	not	[44]
*Cryptosporidium ornithophilus*	ostrich	not	[45]
*Cryptosporidium ratti*	rats	not	[46]
*Cryptosporidium erinacei*	hedgehogs	yes	[41,47]
*Cryptosporidium sciurinum*	red squirrels	not	[48]
*Cryptosporidium myocastoris*	nutria	not	[49]
*Cryptosporidium testudinis*	tortoises	not	[50]
*Cryptosporidium avium*	birds	not	[51]
*Cryptosporidium alticolis*	common voles	not	[52]
*Cryptosporidium microti*	common voles	not	[52]
*Cryptosporidium abrahamseni*	fish	not	[53]
*Cryptosporidium bollandi*	fish	not	[54]
*Cryptosporidium apodemi*	rats	not	[55]
*Cryptosporidium ditrichi*	rodents	yes	[55,56]
*Cryptosporidium ducismarci*	tortoises	not	[50]
*Cryptosporidium proliferans*	rodents	not	[57]

**Table 2 pathogens-11-00515-t002:** Use of drugs for the treatment of human cryptosporidiosis.

Treatment	Clinical Improvement	Protocols	Author(s)
Nitazoxanide	inhibits oocyst excretion	clinical trials	[90]
Paromomycin	no clinical benefits	clinical trials	[91]
Spiramycin	oocyst reduction	in vitro and in vivo studies	[92]
Macrolides (Azithromycin, Erythromycin, Roxithromycin)	no clear clinical benefits	in vitro and in vivo studies	[93]
Clofazimine	oocyst reduction	clinical trials	[94]
Benzoxaboroles, Pyrazolopyridine	oocyst reduction	in vivo studies	[95]

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
