# Peer review of "Cryptosporidium: Still Open Scenarios"

_pathogens, 2022, doi:10.3390/pathogens11050515_

Round 1
Reviewer 1 Report
The manuscript by Pane and Putignani reviewed the recent study advances on Cryptosporidium epidemiology, drug treatment, and culture models, as well as the transmission risk factors for human cryptosporidiosis. Overall, the manuscript is well written and the data are interesting.
Line 59. Should be “the 18s rRNA/gp60 gene sequencing”
Line 77. “C parvum” should be “C. parvum”
Line 78. Only some Cryptosporidium species are absent for host specificity, notably C. parvum.
Lines 84-87 and 99-100. Combine these sentences since they all concern water-related descriptions. There are also similar descriptions distributed separately from lines 81 to 116, please rearrange them.
Line 117. Please add more information on Cryptosporidium genotyping and subtyping tools and discuss their advantages and disadvantages in this section.
Table 1. Why were only some of Cryptosporidium species included here? Please explain. Is it better to present all the human-pathogenic Cryptosporidium spp. reported thus far and their rare or common hosts here?
Figure 1. Please format the contents in the boxes in Figure 1, the alignment is different among them.
Author Response
The manuscript by Pane and Putignani reviewed the recent study advances on Cryptosporidium epidemiology, drug treatment, and culture models, as well as the transmission risk factors for human cryptosporidiosis. Overall, the manuscript is well written and the data are interesting.
Response: we would thank you for the appreciation to our manuscript.
Point 1: Line 59. Should be “the 18s rRNA/gp60 gene sequencing”.
Response 1: Done. We have modified the sentence accordingly to your suggestion.
Point 2: Line 77. “C parvum” should be “C. parvum”
Response 2: Done. We have corrected the word at line 82.
Point 3: Line 78. Only some Cryptosporidium species are absent for host specificity, notably C. parvum
Response 3: Done. We have modified the sentence, accordingly to your suggestion, from to line 83 to line 84.
Point 4: Lines 84-87 and 99-100. Combine these sentences since they all concern water-related descriptions. There are also similar descriptions distributed separately from lines 81 to 116, please rearrange them.
Response 4: Done. We have rearranged sentences accordingly to your suggestion.
Point 5: Line 117. Please add more information on Cryptosporidium genotyping and subtyping tools and discuss their advantages and disadvantages in this section.
Response 5: Done. We have expanded the section with two studies describing advantages and disadvantages of the genotyping and subtyping methodology (10.1016/j.biotechadv.2008.02 .003; 10.1016/j.actatropica.2017.10.023). Please see from line 164.
Point 6: Table 1. Why were only some of Cryptosporidium species included here? Please explain. Is it better to present all the human-pathogenic Cryptosporidium spp. reported thus far and their rare or common hosts here?
Response 6: Done. The Table has been extended including up to 44 species, listed including their rare or common hosts.
Point 7: Figure 1. Please format the contents in the boxes in Figure 1, the alignment is different among them.
Response 7: Done. The format has been revised consistently with your request.
Reviewer 2 Report
The review by Pane and Putignani includes an overview about the epidemiology of Cryptosporidium spp. including the molecular diversity of the most prevalent species of this genus (C. hominis and C. parvum). In addition, Authors comment the available treatments for human cryptosporidiosis and the alternative models to identify new targets for the development of new drugs and to understand the host-parasite interactions. Even though Authors provide relevant data about Cryptosporidium spp., several issues need to be addressed.
Major issues
- Epidemiology section (Table 1): There are some issues. For instance, hominis and C. parvum are repeated in Table 1. In the case of C. hominis, the major host is human. Regarding C. wrairi, Authors comment that pigs are the major host; however, guinea pig is the main reservoir host of this species of Cryptosporidium (see Vetterling et al. J protozool. 1971; 18(2):243-7). Please amend.
- Epidemiology section. Authors mention that 38 species of Cryptosporidium have been described, but up to now, at least 44 species and 120 genotypes of Cryptosporidium have been reported (see Ryan et al. Animals (Basel);2021; 11(11):3307).
- Epidemiology section: Authors do not comment the number of Cryptosporidium species reported in humans. To date, 19 Cryptosporidium species have been identified in humans (see Ryan et al. Animals (Basel);2021; 11(11):3307) . This information can be presented in Table 1.
- Epidemiology section. I miss a paragraph devoted to the role of domestic and wild animals in the transmission of Cryptosporidium This part is not/or poorly described in the manuscript. Cryptosporidium spp. is an important zoonotic parasite, for this reason, a paragraph devoted to the epidemiology in animals should be added to this section.
- Epidemiology section (Lines 103- 113): Please include the type population where this transmission was reported (immunocompetent individuals, immunocompromised patients, both populations)
- Genotyping and spreading section. Although Authors comment the molecular diversity of C. hominis and C. parvum, the genetic diversity of other Cryptosporidium species that can infect humans is not commented (C. melagridis, C. canis, C. felis…) Please include the dominant families/genotypes of these species identified in humans and animals.
- Genotyping and spreading section. For which species is available gp60 marker?
- Cryptosporidium treatments section. This section is poorly described. The information about Cryptosporidium vaccine is limited. Authors should include more information about this topic.
- Cryptosporidium treatments section. Please add a paragraph devoted to the limitations to develop new drugs for Cryptosporidiosis (see Ryan & Hijjawi. Int J Parasitol. 2015; 45(6):367-73)
- Another topic that Authors should comment in this manuscript is the association between Cryptosporidium and microbiota. For instance the presence of C. parvum increase the abundance of Fusobacterium (see Ichikawa-Seki et al. Sci Rep. 2019; 9(1):12517). Include in this section others cases.
Minor issues
- Line 175. Add spp. after Cryptosporidium
- Lines 186-189. Reference for this statement.
Author Response
The review by Pane and Putignani includes an overview about the epidemiology of Cryptosporidium spp. including the molecular diversity of the most prevalent species of this genus (C. hominis and C. parvum). In addition, Authors comment the available treatments for human cryptosporidiosis and the alternative models to identify new targets for the development of new drugs and to understand the host-parasite interactions. Even though Authors provide relevant data about Cryptosporidium spp., several issues need to be addressed.
Response: we would thank you for the opportunity to improve our manuscript following your suggestions.
Response to Reviewer 2 Comments
Major issues
Point 1: Epidemiology section (Table 1): There are some issues. For instance, hominis and C. parvum are repeated in Table 1. In the case of C. hominis, the major host is human. Regarding C. wrairi, Authors comment that pigs are the major host; however, guinea pig is the main reservoir host of this species of Cryptosporidium (see Vetterling et al. J protozool. 1971; 18(2):243-7). Please amend
Response 1: Done. We have corrected the text accordingly with the comment and we have also introduced the reference 10.1111/j.1550-7408.1971.tb03315.x, as suggested.
Point 2: Epidemiology section. Authors mention that 38 species of Cryptosporidium have been described, but up to now, at least 44 species and 120 genotypes of Cryptosporidium have been reported (see Ryan et al. Animals (Basel);2021; 11(11):3307)
Response 2: Done. Please see the new sentence in which we have amended this point, changing the text from line 70 to line 71. We have also introduced the reference 10.3390/ani11113307, as suggested.
Point 3: Epidemiology section: Authors do not comment the number of Cryptosporidium species reported in humans. To date, 19 Cryptosporidium species have been identified in humans (see Ryan et al. Animals (Basel);2021; 11(11):3307). This information can be presented in Table 1.
Response 3: Done. The new Table 1 includes all Cryptosporidium species identified in humans so far.
Point 4: Epidemiology section. I miss a paragraph devoted to the role of domestic and wild animals in the transmission of Cryptosporidium This part is not/or poorly described in the manuscript. Cryptosporidium spp. is an important zoonotic parasite, for this reason, a paragraph devoted to the epidemiology in animals should be added to this section.
Response 4: We thanks the Reviewer for the comment. Please consider the new paragraph 3 titled “Genotyping and spreading” which has been completely rewritten.
Point 5: Epidemiology section (Lines 103- 113): Please include the type of population where this transmission was reported (immunocompetent individuals, immunocompromised patients, both populations)
Response 5: Done. As suggested, we have corrected the sentence by specifying immunocompromised patients. Please, see from line 111.
Point 6: Genotyping and spreading section. Although Authors comment the molecular diversity of C. hominis and C. parvum, the genetic diversity of other Cryptosporidium species that can infect humans is not commented (C. melagridis, C. canis, C. felis…) Please include the dominant families/genotypes of these species identified in humans and animals
Response 6: Following this suggestion, we have included other Cryptosporidium species that can infect humans and animals to represent its genetic diversity. Please see from line 151 to line 163.
Point 7: Genotyping and spreading section. For which species is available gp60 marker?
Response 7: Following this suggestion, we specified for which Cryptosporidium species the gp60 marker is used. Please see from line 129 to line 132.
Point 8: Cryptosporidium treatments section. This section is poorly described. The information about Cryptosporidium vaccine is limited. Authors should include more information about this topic.
Response 8: Following this suggestion, we have better explained this topic and rewritten the paragraph. Please see from line 238 to line 246.
Point 9: Cryptosporidium treatments section. Please add a paragraph devoted to the limitations to develop new drugs for Cryptosporidiosis.
Response 9: As suggested, we have underlined the difficulties in drug development. Please see from line 238 to line 246.
Point 10: Another topic that Authors should comment in this manuscript is the association between Cryptosporidium and microbiota. For instance the presence of C. parvum increase the abundance of Fusobacterium (see Ichikawa-Seki et al. Sci Rep. 2019; 9(1):12517). Include in this section others cases.
Response 10: We thanks the Reviewer for the comment. We have tried to better specify the modulation of microbial flora during cryptosporidiosis, describing the bacterial populations that predominate over other taxa in the reported paper. Moreover, we have reported also the paper doi:10.1093/cid/ciab207 that discusses the possible role of Megasphaera as marker of diarrhea during cryptosporidiosis. Please see from line 257 to line 271.
Minor issues
Point 1: Line 175. Add spp. after Cryptosporidium.
Response 1: Done. Please see line 206.
Point 2: Lines 186-189. Reference for this statement.
Response 2: Done. Actually, the sentence was wrong. We have amended the sentence in which the word “not” was not correctly reported. Please see the new text at line 219.
Round 2
Reviewer 2 Report
Although Authors have added some of my comments and suggestions to the manuscript, there are important issues that need to be addressed.
Major issues
- Table 1: I suggest modify the table caption. Authors comment that species of Cryptosporidium in table 1 are rarely found in humans, and this statement is not correct because C. hominis and C. parvum are common species identified in humans.
- Following the same issue, in Table 1 there are some errors in the major host column. The major host of C. andersoni, C. suis, C. meleagridis, C. scrofarum, C. bovis, C. tyzzeri, C. viatorum, C. ditrichi is not the human. In addition to this, the major hosts of C. suis are pigs and wild boars. For C. tyzzeri, C. viatorum, C. occultus and C. ditrichi are rodents. Please modify.
- Include in Table 1 a new column indicating if these species have been reported in humans.
- Genotyping and spreading section: As mentioned in my previous review of this manuscript, a paragraph devoted to Cryptosporidium spp. in wildlife should be added in this section. Wildlife can act as reservoir of zoonotic Cryptosporidium species. Add in this section if genotypes identified in wildlife have been reported in humans.
- Cryptosporidium treatments section (Lines 255-269): I suggest to the Authors, separate these two paragraphs into a new section named Microbiota and Cryptosporidium spp. Include in this section more examples about the association between Cryptosporidium and other pathogens (not only bacteria). This information provides valuable insights in the Cryptosporidium field.
Minor issues
- Line 85. Add spp. after Cryptosporidium
- Line 123: Replace “Leary” by “O’ Leary”
Author Response
- Table 1: I suggest modify the table caption. Authors comment that species of Cryptosporidium in table 1 are rarely found in humans, and this statement is not correct because C. hominis and C. parvum are common species identified in humans
Response 1: Done. We have modified the table caption, accordingly to your suggestion.
- Following the same issue, in Table 1 there are some errors in the major host column. The major host of C. andersoni, C. suis, C. meleagridis, C. scrofarum, C. bovis, C. tyzzeri, C. viatorum, C. ditrichi is not the human. In addition to this, the major hosts of C. suis are pigs and wild boars. For C. tyzzeri, C. viatorum, C. occultus and C. ditrichi are rodents. Please modify.
Response 2: Done. We have modified the table, accordingly to your suggestion.
- Include in Table 1 a new column indicating if these species have been reported in humans.
Response 3: Done. We have included in Table 1 a new column, accordingly to your suggestion.
- Genotyping and spreading section: As mentioned in my previous review of this manuscript, a paragraph devoted to Cryptosporidium spp. in wildlife should be added in this section. Wildlife can act as reservoir of zoonotic Cryptosporidium species. Add in this section if genotypes identified in wildlife have been reported in humans.
Response 4: Done. We have expanded the section with studies reporting wildlife as reservoir of zoonotic Cryptosporidium species. Please see from line 268 to line 298.
- Cryptosporidium treatments section (Lines 255-269): I suggest to the Authors, separate these two paragraphs into a new section named Microbiota and Cryptosporidium spp. Include in this section more examples about the association between Cryptosporidium and other pathogens (not only bacteria). This information provides valuable insights in the Cryptosporidium field.
Response 5: Done. We thanks the Reviewer for the comment. Please consider the new paragraph 6 titled “Microbiota and Cryptosporidium spp.”
Minor issues
- Line 85. Add spp. after Cryptosporidium
Response 1: Done.
- Line 123: Replace “Leary” by “O’ Leary”
Response 2: Done.
Round 3
Reviewer 2 Report
The manuscript has notably improved. Authors have included all my suggestions and comments of my review. There are minor aspects that should be revised:
- Line 63. Add "and genotypes" after "species"
- Line 83: Remove "marsupials" because these animals are included in mammals.
Author Response
- Line 63. Add "and genotypes" after "species"
- Response 1: We have modified the sentence, accordingly to your suggestion. See line 64.
- Line 83: Remove "marsupials" because these animals are included in mammals.
- Response 2: We have modified the sentence, accordingly to your suggestion. See line 84.